# Revisiting the *Plasmodium* sporozoite inoculum and elucidating the efficiency with which malaria parasites progress through the mosquito

Sachie Kanatani [1,2,5] ✉, Deborah Stiffler[1,2,5], Teun Bousema [3], Gayane Yenokyan[4] & Photini Sinnis [1,2] ✉

Malaria is initiated when infected anopheline mosquitoes inoculate sporozoites as they probe for blood. It is thought that all infected mosquitoes are equivalent in terms of their infectious potential, with parasite burden having no role in transmission success. In this study, using mosquitoes harboring the entire range of salivary gland sporozoite loads observed in the field, we demonstrate a strong and highly significant correlation between mosquito parasite burden and inoculum size. We then link the inoculum data to oocyst counts, the most commonly-used metric to assess mosquito infection in the field, and determine the efficiency with which oocyst sporozoites enter mosquito salivary glands. Taken together our data support the conclusion that mosquitoes with higher parasite burdens are more likely to initiate infection and contribute to onward transmission. Overall these data may account for some of the unexplained heterogeneity in transmission and enable more precise benchmarks for transmission-blocking interventions.

Malaria remains one of the most important infectious diseases globally, responsible for over 600,000 deaths and hundreds of millions of clinical cases per year[1]. Parasites of the genus *Plasmodium* are the causative agents of malaria and are transmitted to humans by infected *Anopheles species* mosquitoes. The parasite cycles between its mammalian and mosquito hosts using specialized developmental stages, gametocytes and sporozoites, and experiences significant bottlenecks as it moves between its obligate hosts, making these stages critical targets for elimination efforts.

Mosquito infection begins with the ingestion of gametocytes. Transformation to gametes followed by sexual reproduction of the parasite occurs in the mosquito midgut ultimately generating motile ookinetes that come to rest on the midgut apical surface and transform to oocysts. The oocyst is the expansion phase of the parasite in the mosquito, with the generation of thousands of sporozoites from a single oocyst. When mature, sporozoites are released into the hemolymph and enter the salivary glands, where they wait to be inoculated into a mammalian host. As infected mosquitoes probe for blood, they inoculate motile sporozoites into the skin of the mammalian host and must find and enter blood vessels to be carried to the liver, where they invade hepatocytes and initiate the first expansion phase in the mammalian host, the exo-erythrocytic form (EEF). Each EEF produces 5000 to 10,000 hepatic merozoites that initiate blood-stage infection, ultimately leading to the production of gametocytes that will be taken up by mosquitoes to continue the life cycle.

A central component of the basic reproductive rate, $R_0$, for malaria is the likelihood that an infected mosquito will initiate infection in a susceptible host[2]. To date, this probability has been estimated

[1]Department of Molecular Microbiology & Immunology, Johns Hopkins Bloomberg School of Public Health, Baltimore, MD, USA. [2]Johns Hopkins Malaria Institute, Johns Hopkins Bloomberg School of Public Health, Baltimore, MD, USA. [3]Department of Medical Microbiology & Radboud Center for Infectious Diseases, Radboud University Medical Center, Nijmegen, the Netherlands. [4]Department of Biostatistics, Johns Hopkins Bloomberg School of Public Health, Baltimore, MD, USA. [5]These authors contributed equally: Sachie Kanatani, Deborah Stiffler. ✉e-mail: skanata1@jhu.edu; psinnis1@jhu.edu

using the entomological inoculation rate (EIR), the number of infected bites per person per unit time. Although EIR correlates with transmission intensity, it lacks the granularity necessary to make robust predictions about the efficacy of transmission-blocking strategies and generate models that explain local heterogeneity in transmission dynamics[3]. Using the biting rate to estimate transmission success assumes that all infected mosquitoes are equally likely to initiate infection, although we know from both controlled human malaria infections (CHMI) and from indirect field estimates, that the majority of bites do not result in infection[4–6]. If biting success were random, EIR would be a reasonable estimate of the success of an infected mosquito bite. However, if there is heterogeneity in the infected mosquito population, such that some mosquitoes were more likely to transmit than others, a better understanding of these factors would improve our epidemiological models and change the benchmarks for transmission-blocking interventions.

Our recent study in the rodent model *Plasmodium yoelii*, demonstrates that mosquito parasite burden is an important determinant of infection likelihood, challenging the assumption that any infected mosquito is an infectious mosquito[7]. Mosquitoes with greater than 10,000 to 20,000 salivary gland sporozoites were 7.5 times more likely to initiate a blood-stage infection compared to mosquitoes with lower parasite burdens. Furthermore, the relationship between infection likelihood and mosquito parasite burden is best described by a threshold model, with a rapid rise in infection probability when mice are bitten by mosquitoes harboring greater than 10,000 to 20,000 sporozoites. Although it is somewhat intuitive that mosquito parasite burden would have a role in transmission success, this has not been the working hypothesis of the field. Indeed, it has been assumed that all infected mosquitoes are equally likely to initiate infection[8].

The reasons for this go back to two previous lines of investigation: The demonstration that mosquito sporozoite load, i.e., infection intensity, and sporozoite inoculum are not correlated[9–12] and the finding that few sporozoites inoculated intravenously could initiate infection with both human and rodent malaria parasites[13,14]. Thus, since inoculum was not dependent upon gland load and few sporozoites could initiate infection, it was concluded that salivary gland sporozoite load was a poor indicator of transmission potential[15]. Indeed, a 2012 review on the topic concluded, "in calculating the EIR it has been assumed that the presence (and not the number) of sporozoites in the salivary glands of a mosquito equates to the mosquito being infectious"[8].

In this study, we revisit the relationship between sporozoite inoculum and mosquito sporozoite load. Following this, we connect salivary gland loads to oocyst numbers, defining the core quantitative relationships between successive parasite stages in the mosquito, in both human and rodent malaria parasites.

## Results

### *Plasmodium falciparum* salivary gland sporozoite loads correlate with inoculum size

To determine the impact of salivary gland sporozoite load on inoculum size of *P. falciparum* infected mosquitoes, we performed single mosquito feeds on the ears of mice using mosquitoes with a range of infection intensities. In order to cover the entire 4 to 5 log-range of observed salivary gland sporozoite loads[9,16–18], we fed mosquitoes on gametocyte cultures (NF54 strain) of 0.03% or 0.3% gametocytemia, resulting in high prevalence infections (81% and 95% respectively), a median infection intensity of 1.5 (IQR: 0.8–4.3) and 22 (IQR: 8–39) oocysts, respectively (Supplementary Fig. 1a), and a broad distribution of salivary gland sporozoite loads (Supplementary Fig. 1b). Since salivary gland loads of 10,000 to 20,000 sporozoites are an inflection point for increased infection probability in the rodent model[7], we made sure to have adequate numbers of mosquitoes above and below this range by periodically assessing any gaps in our data and adjusting

the gametocyte cultures accordingly. Twenty independent batches of infected mosquitoes were generated and provided the necessary spread in sporozoite loads.

Individual mosquitoes from each batch were placed in feeding tubes, starved overnight to increase the likelihood of feeding, and given access to a restricted portion of the ear of an anaesthetized mouse for 10 to 15 min. The amount of time each mosquito spent probing was recorded and after the allotted time, salivary glands were harvested from the mosquito and the relevant portion of the mouse ear was removed for gDNA isolation and sporozoite quantification by qPCR (experimental outline shown in Fig. 1a). Using oligonucleotide primers specific for a large ribosomal subunit fragment (LSUE) encoded in the *Plasmodium* mitochondrial genome and standards made with known numbers of salivary gland sporozoites mixed with ear tissue, we found the limit of quantification (LOQ) was 10 sporozoites (Supplementary Fig. 2a).

In our experience, *P. falciparum* salivary gland loads are maximal between days 14 and 20. We quantified inocula of mosquitoes on both ends of this 7-day range (d14-16 and d18-20) to determine if there was a difference and did not observe significant differences in inoculum size (Supplementary Fig. 3) and thus, pooled the data. The results of 126 individual mosquito-mouse feeds demonstrate a strong correlation between mosquito salivary gland sporozoite load and inoculum size (Fig. 1b, c; Spearman correlation, $\rho = 0.66$, 95% confidence interval 0.55–0.75; $P < 0.0001$). Of the mosquitoes that transmitted sporozoites ($n = 112$), the geometric mean inoculum was 862 sporozoites (IQR: 354–2,666). A small number of mosquitoes ($n = 14$) did not transmit sporozoites with the majority of non-transmitters (86%) having salivary gland loads under 10,000 sporozoites (Fig. 1d). As shown, over half of the mosquitoes ($n = 78$) inoculated >500 sporozoites, with most (89%) of these high inocula from mosquitoes with salivary gland sporozoite loads greater than 10,000. In contrast, of the mosquitoes that inoculated <500 sporozoites ($n = 48$), 65% had salivary gland sporozoite loads less than 10,000. Overall, the odds of inoculating >500 sporozoites is 14 times higher (95% confidence interval 5.2–39.1) among mosquitoes with over 10,000 salivary gland sporozoites compared to mosquitoes harboring less than 10,000 sporozoites. These data differ from previously published studies in two respects: 1) we demonstrate with high confidence and strong statistical significance, a clear relationship between inoculum size and mosquito parasite burden, and 2) the average inoculum size in our study is 10-fold higher than previously reported.

We also examined the transmission efficiency, defined as the percent of total salivary gland load inoculated (inoculum / inoculum plus sporozoites remaining in the glands). As shown in Fig. 1e, ~40% of mosquitoes inoculated between 1 to 5% of the sporozoites in their salivary glands. Not surprisingly, the majority of mosquitoes that did not inoculate sporozoites had low infections (<10,000 salivary gland sporozoites) although a few ($n = 10$) of these low-infected mosquitoes were able to transmit a large percentage of their gland load, accounting for the majority of mosquitoes with transmission efficiencies >20%.

Interestingly, no correlation was found between probe time and salivary gland sporozoite load or sporozoite inoculum size (Supplementary Fig. 4). Although a previous study found that infected mosquitoes probe longer than uninfected mosquitoes[19], we did not find a difference in probe time among infected mosquitoes that were highly infected versus those with low infections, nor did we find that longer probe times correlated with the number of sporozoites inoculated. This is not surprising given the previous demonstration that the majority of sporozoites are inoculated in the first few minutes of probing[20] and our recent work showing an effect of probe time only when large numbers of mosquitoes with short probe times (10 s and 1 min) were included[7]. In our experiments the mean probe time was 480 s (SD ± 200), with 94% of mosquitoes probing for longer than

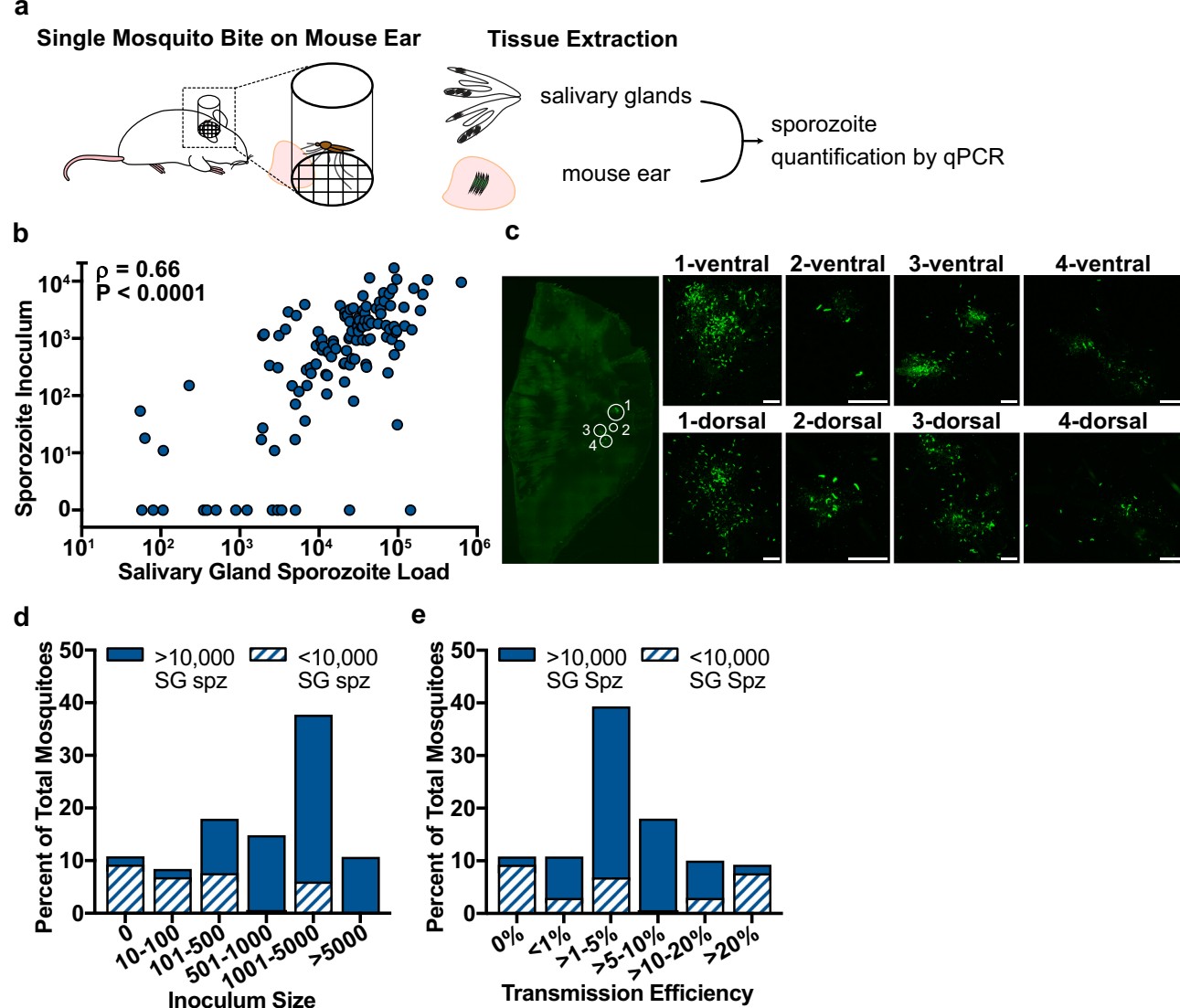

**Fig. 1 | *Plasmodium falciparum* salivary gland sporozoite load correlates significantly with sporozoite inoculum size. a** Schematic of the experimental procedure showing the single mosquito bite set-up followed by the isolation of mosquito salivary glands and mouse ear, and qPCR quantification of sporozoites in these tissues. **b** The sporozoite inoculum positively correlates with mosquito salivary gland sporozoite load: Spearman correlation $\rho = 0.66$, 95% confidence interval 0.55−0.75; $P < 0.0001$ (two-tailed). Each dot represents data from a single mosquito-mouse pair ($n = 126$ from 20 independent experiments). **c** Representative image of CSP-stained sporozoites (green) at a single mosquito bite site from 10 single mosquito bite sites. Left panel is a snapshot of the entire ear with the location of sites at which sporozoites were found delineated by circles. Right panels show zoomed in images of sporozoites at each site. Scale bars: 50 μm. **d** Inoculum size was binned as indicated. Mosquitoes in each bin were classified according to the salivary gland sporozoite load (filled bars >10,000 salivary gland sporozoites and striped bars <10,000 salivary gland sporozoites). Significant differences in inoculum size were observed between mosquitoes with >10,000 and <10,000 salivary gland sporozoites (one-sided $\chi^2$, $P < 0.0001$). **e** Transmission efficiency: The percentage of salivary gland sporozoites injected during probing (inoculum/inoculum plus residual salivary gland sporozoites) was calculated for each mosquito-mouse pair and then binned as indicated. Mosquitoes in each bin were classified according to their salivary gland sporozoite loads (filled bars >10,000 sporozoites and striped bars <10,000 sporozoites). Significant differences in transmission efficiency were observed between mosquitoes with >10,000 salivary gland sporozoites compared to those with <10,000 sporozoites (one-sided $\chi^2$, $P < 0.0001$). SG Spz salivary gland sporozoites.

3 min, thus, the lack of correlation between inoculum size and probe time in our study is not surprising.

### *Plasmodium falciparum* oocyst sporozoites efficiently colonize salivary glands

Having established that salivary gland load is correlated with inoculum size, we wanted to calibrate these loads to oocyst numbers, the life cycle stage that gives rise to salivary gland sporozoites. Oocysts are the most widely used parameter to assess mosquito infection and determine the efficacy of transmission-blocking strategies. While many studies have demonstrated a correlation between oocyst number and salivary gland sporozoite loads, the efficiency with which the parasite makes this transition has been difficult to accurately assess due to limitations in methodology. Previous studies used a batch approach in which a selection of mosquitoes from a single cage is sampled early to count oocysts and another selection is sampled later to count salivary gland sporozoites[15,21–24]. To overcome the limitations of a batch approach, we quantified these parameters in individual mosquitoes, visualizing ruptured oocysts in the midgut by immunofluorescence followed by confocal microscopy and qPCR quantification of salivary gland sporozoites in the same mosquito (Fig. 2a, b). To distinguish ruptured from unruptured oocysts we built upon recent work of the Bousema group[25]. Using an antibody specific for the circumsporozoite protein (CSP), we validated the approach with tdTomato-expressing *P.*

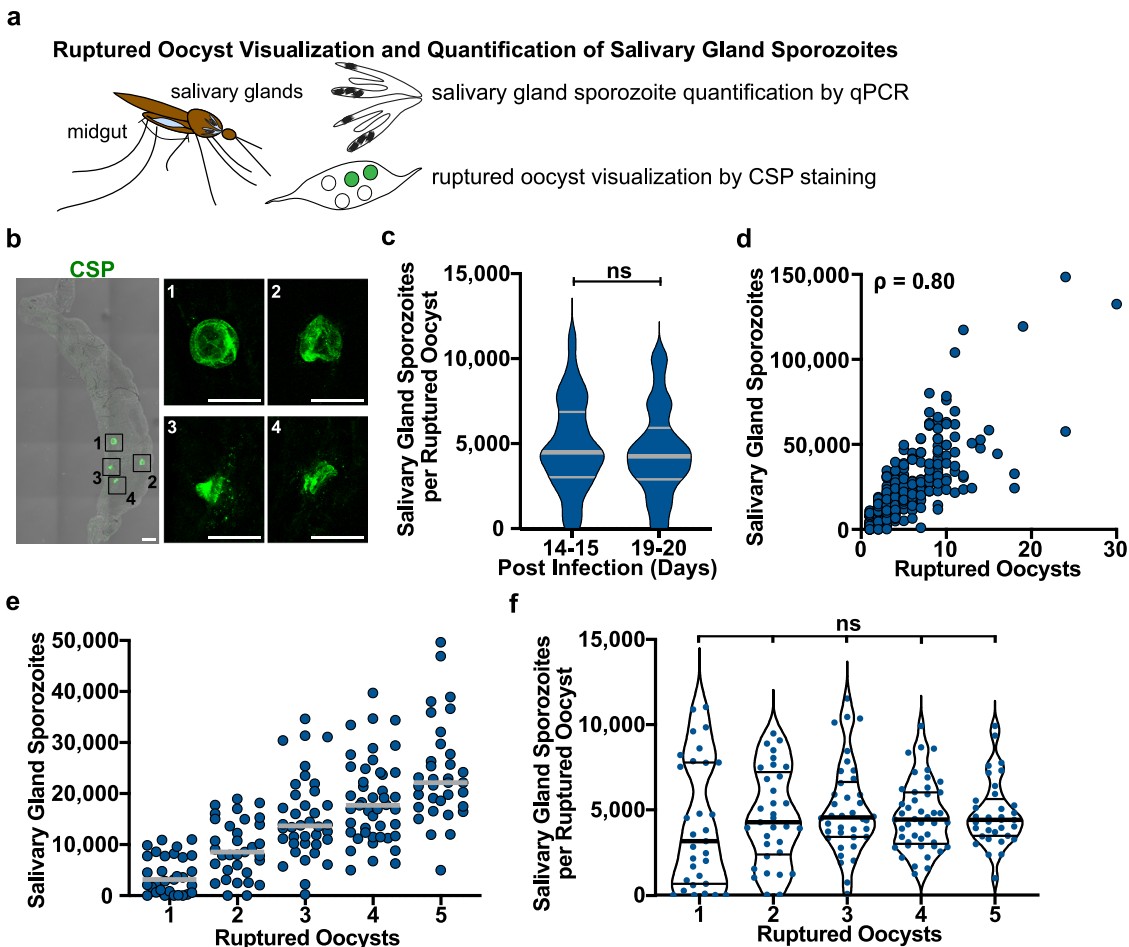

**Fig. 2 | Low numbers of *P. falciparum* ruptured oocysts give rise to high salivary gland sporozoite loads. a** Schematic of the experimental sequence visualizing ruptured oocysts and quantifying salivary gland sporozoites in single mosquitoes performed on days 14–15 and 19–20 post-mosquito infection. **b** Image of CSP-stained ruptured oocysts (green) on a single midgut (representative of the 291 midguts analyzed). Scale bar: 100 μm. Insets show individual ruptured oocysts. Scale bars: 50 μm. **c** Violin plots of salivary gland sporozoites per ruptured oocyst at indicated post-infection days. The thick bar indicates the median, and the thin bars indicate the upper and lower quartiles. Days 14–15 and 19–20 post-infection were compared. (ns, not significant, *P* > 0.1 [two-tailed Mann–Whitney test]). **d** Salivary

gland sporozoite quantity correlates with number of ruptured oocysts: Spearman correlation, $\rho$ = 0.80; *P* < 0.0001 (two-tailed). Each dot represents one mosquito (total *n* = 291 from 10 independent experiments). **e** Dot plots of salivary gland sporozoite numbers for each of the indicated number of ruptured oocysts using the dataset in panel **d**. Each dot represents one mosquito, with bars indicating the median. **f** Violin plots illustrating salivary gland sporozoites per ruptured oocyst using the dataset in panel **e**. The thick bar indicates the median, and the thin bars indicate the upper and lower quartiles. The quantities in each ruptured oocyst were compared to each other (ns not significant, *P* > 0.1 [Kruskal–Wallis test followed by Dunn's test]).

*falciparum* and confirmed that the CSP signal was only observed in ruptured oocysts and did not stain unruptured oocysts (tdTomato signal) (Supplementary Fig. 5a).

We performed these studies on days 14–15 and 19–20 post-infection since at these times over 80% of oocysts have ruptured (Supplementary Fig. 5b) and salivary gland loads are close to maximal. Indeed, when comparing data from days 14–15 and 19–20 post-infection, there were no significant differences in the number of salivary gland sporozoites per ruptured oocysts (Fig. 2c); thus, we pooled the data for a total of 291 individual mosquito observations, covering a range of mosquito infection intensities while focusing on mosquitoes with low oocyst numbers (Fig. 2d, e). As expected, there was a strong correlation between salivary gland sporozoite load and ruptured oocyst numbers (Spearman correlation $\rho$ = 0.80, *P* < 0.0001; Fig. 2d). However, we found that colonization of salivary glands by oocyst sporozoites is a more efficient process than previously reported[15,21–24]. Focusing on mosquitoes with oocyst loads from 1 to 5, the numbers most frequently observed in the field, we found that 42% of mosquitoes with 2 ruptured oocysts and 83% of mosquitoes with 3 ruptured oocysts had salivary gland sporozoite loads >10,000 (Fig. 2e). Salivary

gland loads >20,000 were achieved by 23% of mosquitoes with 3 ruptured oocysts and by over 60% of mosquitoes with 5 or more ruptured oocysts. The median number of salivary gland sporozoites per ruptured oocyst was 4427, and this did not vary significantly with oocyst number (Fig. 2f).

Since few studies have investigated the productivity of single *Plasmodium* oocysts, we next quantified the number of *P. falciparum* sporozoites in mature oocysts. Using mosquitoes with low intensity infections to ensure complete development of oocysts, we stained midguts with mercurochrome and measured their size (Fig. 3a, b). We found that oocyst diameter significantly increased between days 7 to 11 post-infection, and reached a stable maximum after day 11 post-infection (Fig. 3c), with mean diameters of 51.2 μm and 52.6 μm on days 11 and 12 post-infection, respectively. Since oocysts mature asynchronously, we isolated single oocysts from the midgut on days 11 and 12 post-infection by microscopy-guided dissection of the relevant portion of the midgut, measured their diameter and then quantified sporozoite number by qPCR (Fig. 3d, e). Although median oocyst diameter does not change after this point, there is a wide range of oocyst sizes (mean diameter 48.7 μm with a SD ± 8.9 μm),

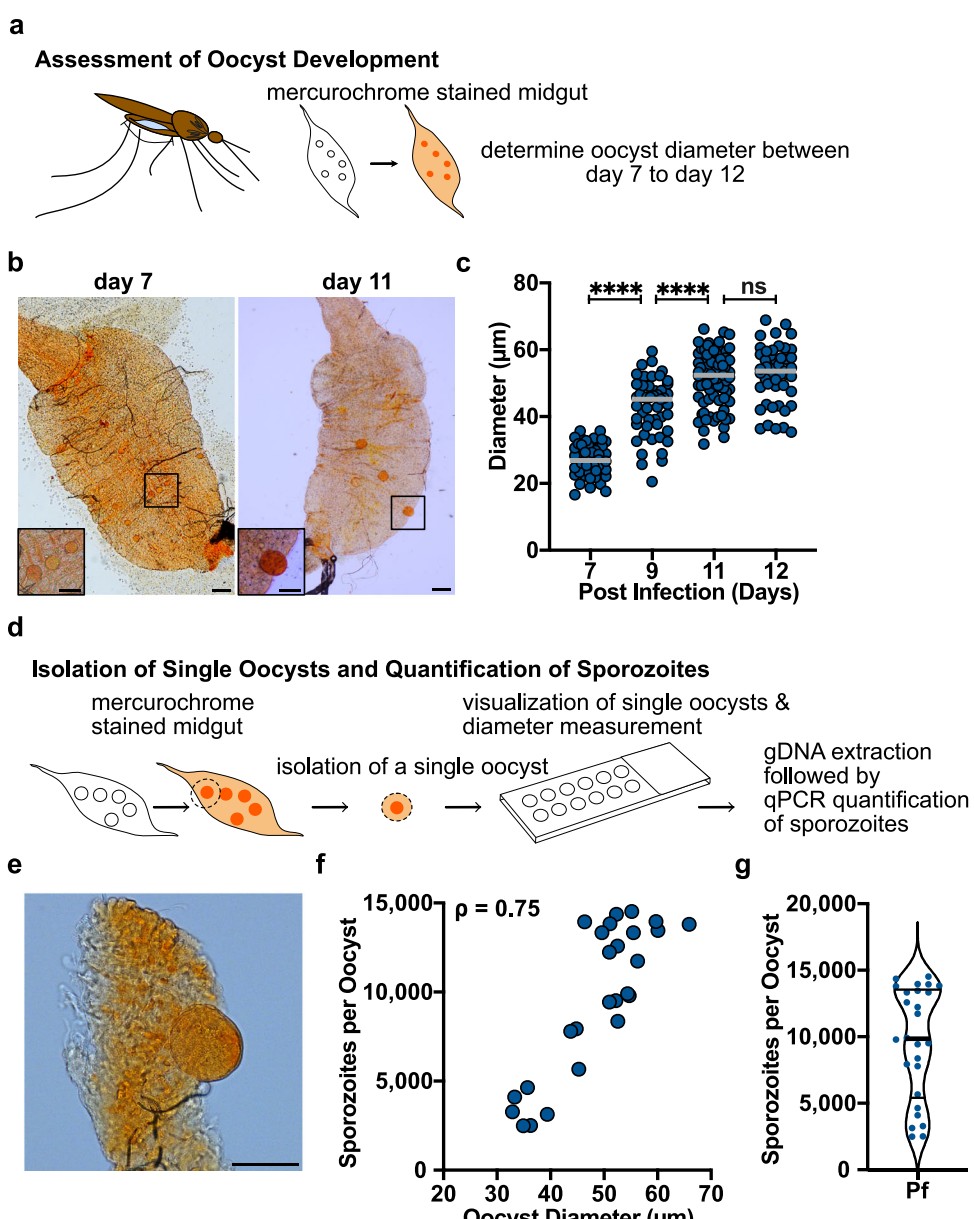

**Fig. 3 | Individual *P. falciparum* oocysts contain a large number of sporozoites.** **a** Schematic of oocyst development assessment. The midguts of *P. falciparum* infected mosquitoes were dissected and stained with mercurochrome between days 7 to 12 post infection. **b** Representative images of *P. falciparum* infected midguts on days 7 and 11 post infection. Scale bars: 100 μm. Insets show zoomed in images of oocysts. Scale bars: 50 μm. **c** *P. falciparum* oocyst diameters on days 7–12 post infection. Each dot represents one oocyst and bars indicate the mean (****$P < 0.0001$; ns, not significant, $P > 0.5$ [one-way ANOVA followed by Tukey's test]). Data are from two independent mosquito cycles and the numbers of midguts analyzed are day 7, $n = 19$; day 9, $n = 20$; day 11, $n = 27$; day 12, $n = 16$) **d** Schematic of sporozoite quantification in single oocysts. On days 11 and 12 post-infection, *P.*

*falciparum*-infected mosquito midguts were stained with mercurochrome to identify oocysts. Single oocysts were isolated by microscopy-guided dissection, placed on slides for diameter measurements and then transferred to a tube for sporozoite quantification by qPCR. **e** Representative image of an isolated single oocyst. Scale bar: 50 μm. **f** Sporozoite quantity correlates with oocyst diameter: Spearman correlation, $\rho = 0.75$; $P < 0.0001$ (two-tailed). Each dot represents a single oocyst (data obtained from three independent mosquito cycles, total $n = 26$). **g** Violin plot illustrating sporozoite quantity in single oocysts using the dataset in panel **f**. The thick bar indicates the median, and the thin bars indicate the upper and lower quartiles.

likely reflecting the biological diversity inherent in this system as well as the asynchronous maturation of oocysts. Given that we cannot accurately measure oocyst diameter prior to their isolation from the midgut, the oocysts we sampled reflected the range of sizes found on days 11 and 12. Within this range, we observed a strong correlation between oocyst diameter and fecundity, where fecundity is defined as the number of individual sporozoites produced by a single oocyst (Spearman correlation, $\rho = 0.75$; $P < 0.0001$; Fig. 3f), with individual oocysts producing between 2489 and 14,520 sporozoites (mean 9604 ± 4215; Fig. 3f). Thus, the fecundity of *P. falciparum* oocysts is

likely a significant contributor to the large number of salivary gland sporozoites observed in mosquitoes with low oocyst numbers. Taken together, our data demonstrate that *P. falciparum* oocyst sporozoites successfully colonize salivary glands with an efficiency of ~48%.

**Comparing the progression of rodent and human malaria parasites through the mosquito**
Since the rodent malaria model is commonly used to assess the efficiency of parasite transitions in the mosquito[12,24,26], it is important to

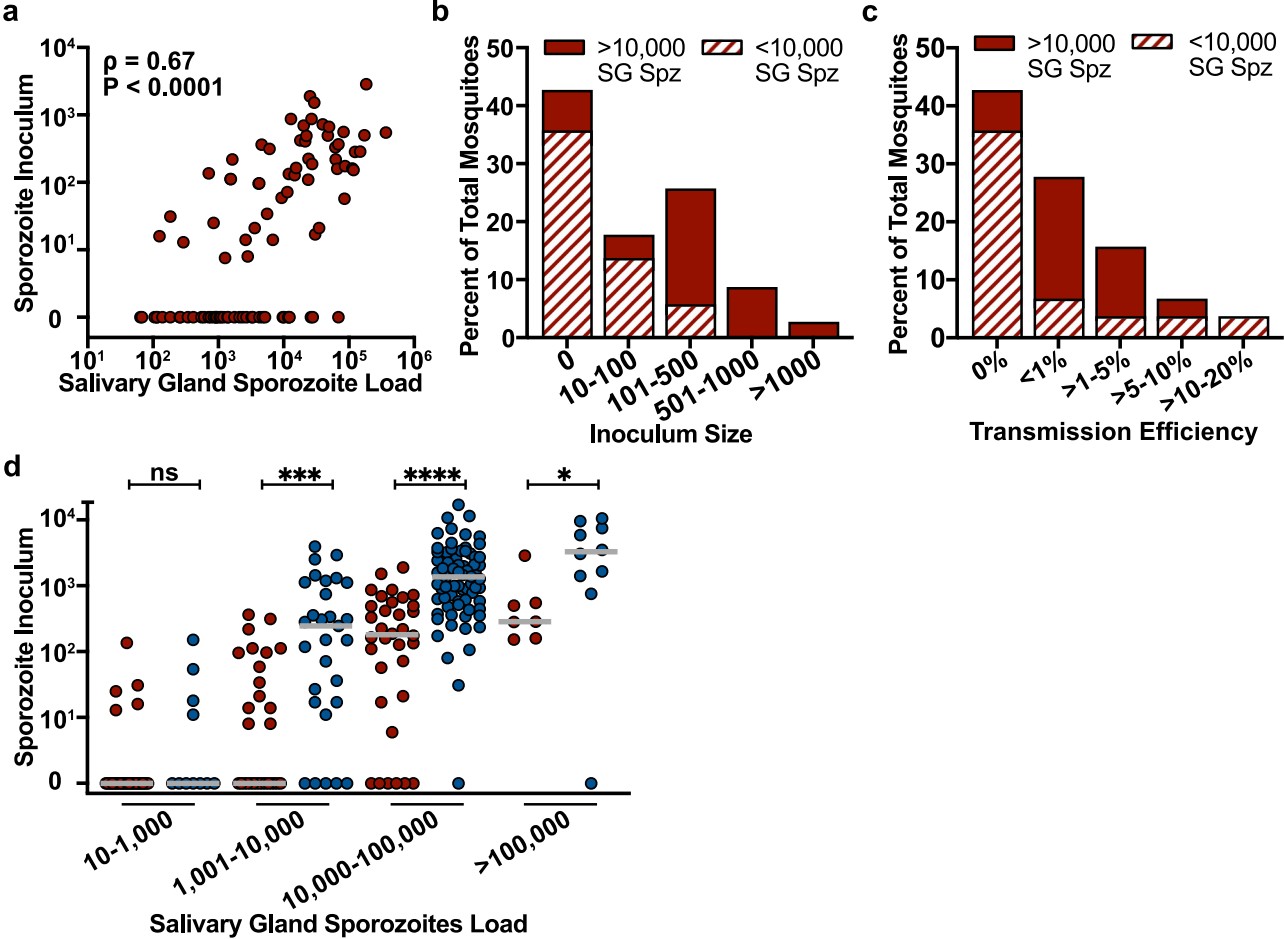

**Fig. 4 | Salivary gland sporozoite load correlates with inoculum size in the rodent malaria model. a** The sporozoite inoculum positively correlates with mosquito salivary gland sporozoite load: Spearman correlation, $\rho = 0.67$, 95% confidence interval 0.55–0.78; $P < 0.0001$ (two-tailed). Each dot represents data from a single mosquito ($n = 94$ from 13 independent experiments). **b** Inoculum size was binned as indicated and mosquitoes in each bin were classified according to their salivary gland sporozoite load (filled bars >10,000 salivary glands sporozoites and striped bars <10,000 salivary glands sporozoites). Significant differences in inoculum size were observed between mosquitoes with >10,000 and <10,000 salivary gland sporozoites (one-sided $\chi^2$, $P < 0.0001$). **c** Transmission efficiency: The percentage of salivary gland sporozoites injected during probing (inoculum/inoculum plus residual salivary gland sporozoites) was calculated for each mosquito-mouse pair and then binned as indicated. Mosquitoes in each bin were classified according to their salivary gland sporozoite load (filled bars >10,000 sporozoites and striped bars <10,000 sporozoites). Significant differences were observed between mosquitoes with >10,000 and <10,000 salivary gland sporozoites (one-sided $\chi^2$, $P < 0.0001$). **d** Comparison of inoculum size of log-binned *P. falciparum* (blue) and *P. yoelii* (red) salivary gland loads (*$P = 0.0136$; ***$P = 0.0001$; ****$P < 0.0001$; ns, not significant, $P > 0.1$ [two-tailed Mann–Whitney test]). Bars indicate median values. SG Spz salivary gland sporozoites.

understand the fidelity with which this model replicates the *P. falciparum* transitions. We chose *P. yoelii* because it would enable us to connect infection likelihood from our previous study[7] with inoculum size and oocyst number, core quantitative relationships that have yet to be robustly defined. To this end, we generated *P. yoelii* infected mosquitoes covering a 4 to 5 log-range of infection intensities. This was accomplished by allowing different batches of mosquitoes to feed on mice with gametocytemias ranging from 0.2% and 0.8% (Supplementary Fig. 1a).

Single-mosquito inoculum studies were performed as outlined for *P. falciparum* and similarly, we observed a strong and highly significant correlation between sporozoite inoculum and salivary gland sporozoite loads (Spearman correlation $\rho = 0.67$, 95% confidence interval 0.55–0.78; $P < 0.0001$) (Fig. 4a). However, in contrast to *P. falciparum*, *P. yoelii*-infected mosquitoes inoculated lower numbers of sporozoites. Of the mosquitoes inoculating sporozoites ($n = 53$), the geometric mean inoculum was 148 (IQR: 58-453) with 18% of these mosquitoes inoculating 10 to 100 sporozoites, 27% inoculating 100 to 500 sporozoites, and 12% inoculating over 500 sporozoites (Fig. 4b). Importantly, a much larger percentage of *P. yoelii*-infected mosquitoes

(44%) did not inoculate sporozoites compared to *P. falciparum* (11%) (Fig. 4c). This is likely due to the larger numbers of low-infected *P. yoelii* mosquitoes (Supplementary Fig. 1b), with 25% of *P. yoelii* infected mosquitoes having ≤ 1000 salivary gland sporozoites compared to only 11% of *P. falciparum* infected mosquitoes. Nonetheless, when inocula of mosquitoes with similar salivary gland loads are directly compared, it is clear that *P. falciparum* inoculate higher numbers of sporozoites (Fig. 4d).

We then performed experiments to link the salivary gland load and inoculum data to oocyst numbers, generating single mosquito data as we did for *P. falciparum*. Although ruptured oocyst number correlated strongly with salivary gland sporozoite loads (Fig. 5a), oocyst sporozoite colonization of salivary glands was significantly less efficient in *P. yoelii*-infected mosquitoes compared to *P. falciparum*. Indeed, when mosquitoes had fewer than 20 ruptured oocysts, salivary gland sporozoite loads over 10,000 were only observed 13% of the time, and greater than 20 ruptured oocysts was required for the majority of mosquitoes have gland loads >10,000 (Fig. 5b). Overall the number of oocyst sporozoites colonizing the salivary glands was significantly lower for *P. yoelii*, with a median of 176 (IQR 76–421) salivary

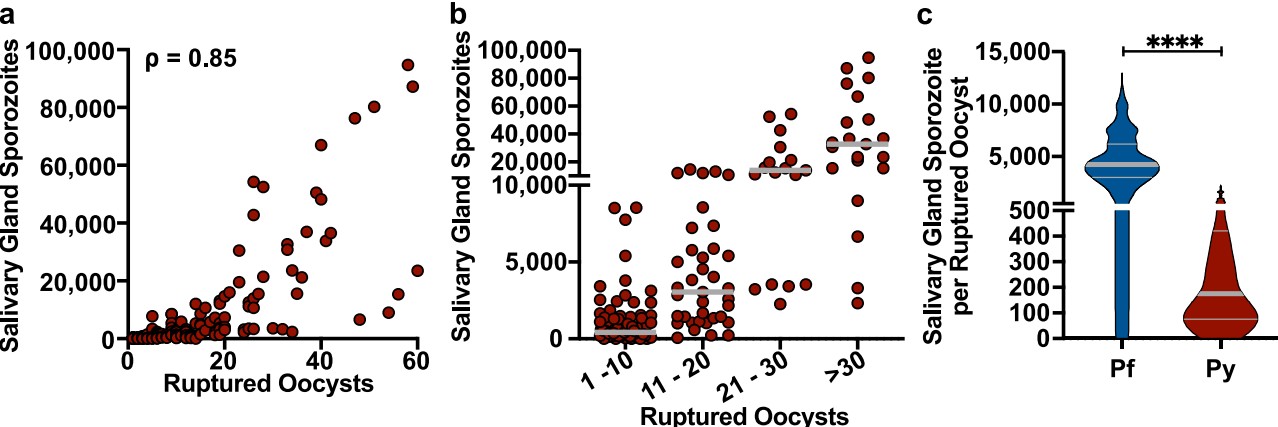

**Fig. 5 | Efficiency with which oocyst sporozoites colonize salivary glands is significantly lower in the rodent model. a** Salivary gland sporozoite load correlates with the number of ruptured oocysts: Spearman correlation, $\rho = 0.85$; $P < 0.0001$ (two-tailed). Each dot represents one mosquito ($n = 158$ from 10 independent experiments). **b** Dot plots illustrating salivary gland sporozoite numbers for the indicated range of ruptured oocysts using the dataset in panel **a**. Each dot represents one mosquito with bars indicating the median. **c** Comparison of salivary gland sporozoite numbers per ruptured oocyst in *P. falciparum* and *P. yoelii* infected mosquitoes. The thick bar indicates the median, and the thin bars indicate the upper and lower quartiles. For *P. falciparum*, the data from Fig. 2d was used; ($P < 0.0001$ two-tailed Mann–Whitney test).

gland sporozoites per ruptured oocyst while for *P. falciparum* it was 4,236 (IQR 2997–11,542; Fig. 5c).

The significant differences in the efficiency with which rodent and human malaria parasites colonize the salivary glands may be due in part to the productivity of individual oocysts. To test this we quantified the size and productivity of mature *P. yoelii* oocysts in the same manner as we did for *P. falciparum*. Monitoring the development of *P. yoelii* oocysts over time we found that growth reached a stable maximum between days 7 to 9 post-infection (Fig. 6a). Interestingly, the size of *P. yoelii* oocysts was similar to the size range observed with *P. falciparum* (Fig. 6b). On days 8 to 9 post-infection, we harvested individual oocysts and quantified the number of sporozoites they harbored by qPCR. Similar to *P. falciparum*, the number of sporozoites in each oocyst correlated with the oocyst diameter (Spearman correlation $\rho = 0.62$, $P = 0.0001$; Fig. 6c). However, *P. yoelii* oocysts were less productive (Fig. 6d), with a mean sporozoite number of 6641 ($\pm$SD of 2438) per oocyst, which is significantly less than what was observed in *P. falciparum* (Mann-Whitney test, $P = 0.0038$; Fig. 6d). Overall, the efficiency with which *P. yoelii* oocyst sporozoites enter salivary glands is 5.3%, which is ~9 times lower than what we observed in *P. falciparum*-infected mosquitoes.

Interestingly, it was originally noted by Garnham that *P. yoelii* salivary gland sporozoites were longer than those of other species[27]. We hypothesized that sporozoite size might be a limiting factor in oocyst productivity and compared the sizes of live, unfixed *P. yoelii* and *P. falciparum* oocyst sporozoites. We found that *P. yoelii* oocysts sporozoites are significantly longer, with a mean length of 13.9 μm, compared to *P. falciparum* where the mean length was 8.9 μm (Unpaired t-test, $P < 0.0001$; Fig. 6e, f). This size difference may allow for more dense packing of *P. falciparum* oocysts sporozoites in both oocysts and salivary glands although that is not likely to be the primary reason behind the lower efficiency of salivary gland invasion by *P. yoelii* sporozoites.

## Discussion

Here, we conclusively demonstrate that mosquito salivary gland sporozoite load correlates with inoculum size with high confidence and strong statistical significance in both human and rodent malaria parasites. This is important because the lack of a correlation between these two variables has been a major reason all infected mosquitoes were thought to be equally likely to initiate infection[8,11,15]. The differences between our study and previous studies can be explained by the

high biological variability inherent in the data and the biased distribution of mosquito salivary gland loads in previous studies. Our recent finding in the rodent model that 10,000 to 20,000 salivary gland sporozoites is an inflection point for infection likelihood, demonstrates the importance of interrogating the entire 4–5 log range of sporozoite salivary gland loads in adequate numbers to overcome the overdispersion in the data. Previous studies were biased towards highly infected mosquitoes[11,12,20] or mosquitoes with low sporozoite loads[10,28], a bias whose importance was not realized at the time. Overall, these data support the conclusion that mosquito parasite burden is an important factor in transmission success.

Inoculum size is critical to an understanding of any infectious disease, impacting the probability of successful transmission as well as the course and severity of disease. In this study, we found that the *P. falciparum* sporozoite inoculum is significantly higher than previously thought. Previous studies with *P. falciparum* in a variety of mosquito hosts (*An. stephensi, An. gambiae, An. freeborni,* and *An. funestus*) found that over 90% of mosquitoes inoculated <50 sporozoites[9,10,20,28]. In contrast, the majority (62%) of *P. falciparum*-infected mosquitoes in this study inoculated >500 sporozoites and almost half (47%) inoculated >1000 sporozoites. Possible explanations for this discrepancy are the low salivary gland sporozoite loads in several of the previous studies[9,10,28] and the use of induced salivation into capillary tubes with immobilized mosquitoes[9,10,20,28] to quantify inocula, which does not allow for normal probing behavior and likely impacts sporozoite expelling. Although one study used methodology more in line with ours, i.e. mosquitoes probing on mouse skin stretched over blood, they observed low incula with a mean 47 ± 96 sporozoites[11]. However, their readout was quantified by fluorescence microscopy, and they removed the subcutaneous tissue under the skin after probing and before quantification, likely leading to a loss of sporozoites. Additional support for the higher than expected inocula we observed is a parallel study from the Bousema group[29] with similar findings and our visualization of a *P. falciparum* bite site with an inoculum consistent with the PCR data. Interestingly, the *P. yoelii* inoculum measurements are similar to previously published studies with rodent malaria parasites[12,30,31]. These experiments were performed with mosquitoes probing on live aneasthetized mice followed by RT-PCR or quantification by microscopy and provide support for the accuracy of the methodology used in our study. Overall, *P. falciparum* inocula are ~5 times higher than what we and others found for the rodent parasites, suggesting that a higher inocula may be needed to successfully infect a

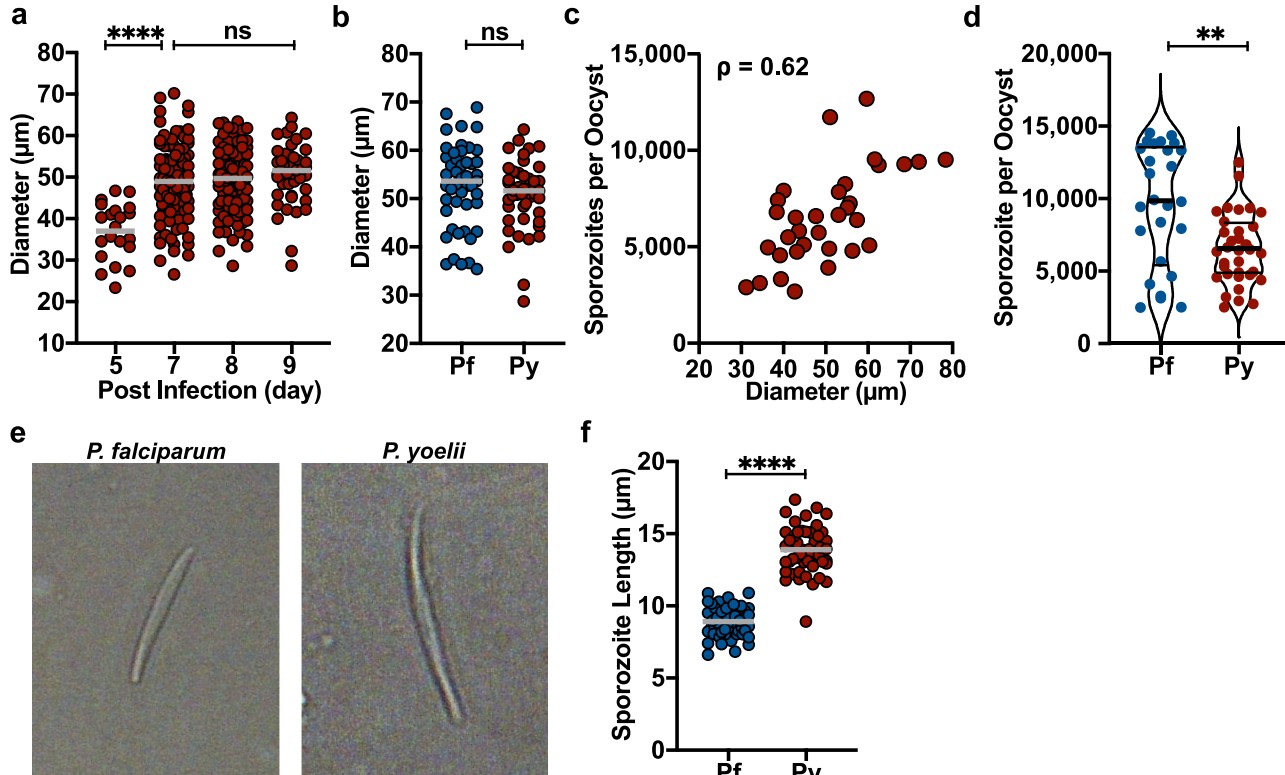

**Fig. 6 | Comparison of oocyst size and productivity in human and rodent malaria parasites. a** *Plasmodium yoelii* oocyst diameter at the indicated days post infection. Each dot represents one oocyst and bars indicate the mean (****$P < 0.0001$; ns, not significant, $P > 0.1$; one-way ANOVA test followed by Tukey's test). Data are from two independent mosquito cycles and the numbers of midguts analyzed are day 5, $n = 11$; day 7, $n = 19$; day 8, $n = 26$; day 9, $n = 17$) **b** Comparison of mature *P. falciparum* (Pf, day 12) and *P. yoelii* (Py, day 9) oocyst sizes. (ns, not significant, $P > 0.1$ [two-tailed unpaired t-test]). **c** Sporozoite quantity correlates with oocyst size in *P. yoelii*: Spearman correlation $\rho = 0.62$; $P < 0.0001$ (two-tailed). Each dot represents a single oocyst ($n = 33$, data obtained from 3 independent mosquito cycles). **d** Violin plots comparing the numbers of sporozoites in single oocysts from *P. falciparum* and *P. yoelii* infected mosquitoes (**$P = 0.0038$; [two-tailed Mann–Whitney test]). Each dot represents a single oocyst and thick bars indicate the median and thin bars indicate the upper and lower quartiles. **e** Representative image of a freshly isolated *P. falciparum* (day 11) and *P. yoelii* (day 9) oocyst sporozoites. Oocyst sporozoites were released by homogenizing infected midguts and placing the released sporozoites on a glass slide for imaging. Scale bar: 5 μm. **f** *P. falciparum* and *P. yoelii* oocyst sporozoite length was plotted and compared (****$P < 0.0001$ [two-tailed unpaired t-test]). Bar indicates the mean ($n = 50$).

larger mammalian host. Nonetheless, replicating these studies in other mosquito species and with additional *P. falciparum* isolates will be informative.

The transition from oocyst to salivary gland sporozoites gives rise to mosquitoes capable of transmission, yet the quantitative dimensions of this transition remain a significant knowledge gap. Filling this gap requires two independent datasets, namely the number of sporozoites that develop in single oocysts and the association between oocyst number and salivary gland sporozoite load. Previous studies investigating the relationship between oocyst numbers and salivary gland sporozoites generated batch averages, which do not consider the asynchronous development of oocysts and the large inter-mosquito variation in oocyst number within a batch of infected mosquitoes. The consensus from these reports was that an average of 500 to 1000 *P. falciparum* and *P. vivax* sporozoites per oocyst successfully colonize salivary glands[21–23,32] while in the rodent model each oocyst gives rise to an average of 70 or fewer salivary gland sporozoites[24,26]. Until now, no studies had quantified oocyst fecundity of rodent malaria parasites and only two studies had been performed with human malaria parasites: Pringle counted 9,555 sporozoites in a single mature *P. falciparum* oocyst[33], while Rosenberg, using a hemocytometer and counting many *P. vivax* and *P. falciparum* oocysts from heavily infected mosquitoes, arrived at an average of 3600 sporozoites per oocyst[32]. Using these data investigators concluded that somewhere between 5 to 20% of oocyst sporozoites reach the salivary glands, with the yield in the rodent model being far lower.

In this study, we found that the efficiency with which *P. falciparum* oocyst sporozoites successfully colonize salivary glands is significantly higher than previously appreciated, with an average efficiency of 48%. Nonetheless, as with many systems in vivo, the data show biological variability and in a minority of mosquitoes the efficiency of salivary gland colonization is higher. This could be explained by oocysts that rupture close to the mosquito's cardia, which could enable sporozoites to catch the hemolymph flow and more efficiently direct them to the glands[34]. It also is possible that in some cases we missed a ruptured oocyst due to its having completely detached from the midgut after rupture and prior to fixation and staining, thus overestimating the salivary gland sporozoite numbers from each oocyst. Although this may have occurred in some instances, the remarkable consistency in salivary gland sporozoites per oocyst over a range of ruptured oocyst counts, suggests our numbers are a good approximation of the efficiency with which oocyst sporozoite colonize the salivary glands. Given the unexpected efficiency with which *P. falciparum* oocyst sporozoites colonize salivary glands, it will be important to replicate these studies in *An. gambiae* and *An. coluzzii*, the primary vectors in sub-Saharan Africa and to determine whether the efficiency with which the parasite moves through the mosquito reflects what is known of their vector competence.

An unexpected finding of our study was the discordance between rodent and human malaria parasites, with several differences noted: *P.yoelii* oocysts produce fewer sporozoites, the oocyst to salivary gland transition is almost 10-fold lower, and *P.yoelii* inocula are lower. While

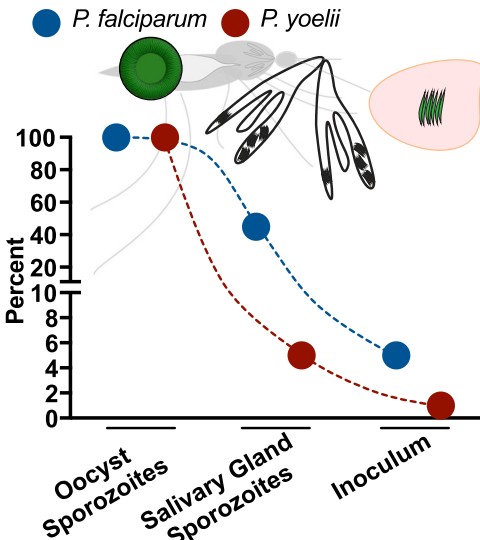

**Fig. 7 | The efficiency with which sporozoites navigate successive hurdles after the parasite's expansion phase in the mosquito.** Shown is the percent of total sporozoites produce by a single *P. falciparum* or *P. yoelii* oocyst that enter salivary glands and are transmitted to the mammalian host.

studies and indirect estimates of infection success in the field found that the majority of infected mosquito bites do not result in blood-stage malaria infection[4–6], suggesting that most inoculated sporozoites do not successfully develop into the next life cycle stage.

Here, we established the core quantitative relationships between successive mosquito stages for *P. falciparum* in *An. stephensi* mosquitoes, resolving critical gaps in our knowledge. A summary of these data is shown in Fig. 7. Our studies reveal a factor that may be responsible for some of the heterogeneity observed in transmission studies[3,39,40]. Previous and recent studies suggest that a minority of mosquitoes harbor 3 or more oocysts and over 10,000 salivary gland sporozoites[41,42]. We hypothesize that this small group of highly infected mosquitoes may be responsible for a disproportionate amount of observed malaria infections and thus explain some of the observed heterogeneity in transmission in the field. It is this group of mosquitoes that are likely to inoculate high numbers of sporozoites that in turn would be expected to result in a successful infection. Although the correlation between salivary gland sporozoite load and inoculum size supports the notion that mosquito parasite burden plays a role in onward transmission, a CHMI trial using single bites from mosquitoes with a range of salivary gland sporozoite loads is needed. This will enable us to define the profile of an infectious mosquito, an outcome that could be incorporated into epidemiological models and could lead to improved benchmarks for transmission-blocking interventions.

## Methods

### Ethics statement

All animal work was conducted in accordance with the recommendations in the Guide for the Care and Use of Laboratory Animals of the National Institutes of Health. The protocol was approved by the Johns Hopkins University Animal Care and Use Committee (Protocol #M020H267), which is fully accredited by Association for Assessment and Accreditation of Laboratory Animal Care. Mice were housed in an environment maintained at 72° Fahrenheit with 42% humidity, where the light cycle is set to turn on at 6:30 AM and off at 9 PM.

### Mosquito infection with *P. falciparum* NF54

*Anopheles stephensi* mosquitoes were infected with *P. falciparum* NF54 as previously described[43]. Briefly, asexual cultures were maintained in vitro in O⁺ erythrocytes at a 4% hematocrit in RPMI 1640 (Corning) supplemented with 74 μM hypoxanthine (Sigma), 0.21% w/v sodium bicarbonate (Sigma), and 10% v/v heat inactivated human serum (Interstate blood bank). Cultures were maintained at 37 °C in a glass candle jar. Gametocyte cultures were initiated when cultures were at 0.5% parasitemia and at 4% hematocrit by changing the media daily for up to 15 to 18 days without the addition of fresh blood to promote gametocytogenesis. Cultures with final gametocytemias of either 0.3% or 0.03% in 40% hematocrit containing fresh O⁺ human serum and O⁺ erythrocytes were used for the mosquito feeding. *Anopheles stephensi* mosquitoes (3-7 days post-emergence) were allowed to feed for up to 30 min. Infected mosquitoes were maintained at 25 °C and 80% humidity and were provided with 10% w/v sucrose solution. Only cages with an infection prevalence above 85% were used for experiments.

### Mosquito infection with *P. yoelii* 17XNL

*Anopheles stephensi* mosquitoes were infected with *P. yoelii* as previously described[44]. Briefly. Swiss Webster mice (Taconic, 5-10 weeks old) were infected with *P. yoelii* 17XNL wild-type parasites and *An. stephensi* mosquitoes (3-7 days post-emergence) were starved for overnight, and then allowed to feed twice on infected mice with gametocytemia between 0.2 and 0.8% for 15 min, with a 6-hour interval between feedings. Three to four days post-infectious blood meal, mosquitoes were provided an additional blood meal from a naïve mouse. Infected mosquitoes were maintained at 24 °C and 80%

the 30% decrease in oocyst fecundity and the 5-fold decrease in inoculum size are not unexpected as the vertebrate host is significantly smaller for the rodent parasites and may require fewer sporozoites to initiate an infection, the inefficiency with which *P. yoelii* oocyst sporozoites transitioned to the salivary gland was surprising. In *P. falciparum*, 3 or more oocysts could reliably lead to salivary gland infections greater than 10,000 sporozoites, while over 20 oocysts were required to yield these gland loads in *P. yoelii*. Indeed, we found that the efficiency with which *P. yoelii* oocyst sporozoites enter salivary glands is ~5%, compared to ~48% in *P. falciparum*. We hypothesize that this is due to some incompatibility between the parasite and its mosquito host. While *An. stephensi* is a natural host for *P. falciparum* parasites, the mosquito host(s) for *P. yoelii* remain unknown[35] and the host for *P. berghei*, *Anopheles dureni millecampsi*[35], cannot be domesticated in the laboratory. The range of infectivity of murine parasites to anopheline mosquitoes has been experimentally tested and many domesticated anophelines are partially or totally refractory to infection with the rodent parasites, with *An. stephensi* being the best experimental vector found to date[26], although even in *An. stephensi*, salivary gland sporozoites loads are comparatively low. Whilst we do not know the distribution of oocyst numbers in the natural mosquito hosts of the murine parasites, if the human parasites are any guide, it would be unusual to require 20+ oocysts to reliably initiate an infection. Thus, we believe that the *An. stephensi*-rodent malaria model does not accurately reflect the oocyst to salivary gland transition that is likely to occur in its natural mosquito host, a fact that should be considered in studies that utilize the rodent model for sporogony and mosquito-host immunity studies.

Despite the inefficiencies of the rodent model, the *P. yoelii* inoculum data can be compared with the infection data of our previous study to estimate the number of sporozoites needed to initiate an infection in the rodent model. Using salivary gland loads from the two studies to relate these variables, we find that an inoculum of ~200 sporozoites is needed to have a high likelihood of initiating infection in the rodent model. This is significantly greater than the number of intravenously inoculated sporozoites required for infection with one study demonstrating that eight IV-inoculated *P. yoelii* sporozoites infected 50% of mice[14]. This discrepancy highlights the bottlenecks that sporozoites face after their inoculation into the skin[36–38] and may be relevant for human malaria parasites. Indeed, CHMI

humidity and were provided with 10% w/v sucrose solution. Only cages with an infection prevalence above 85% were used for experiments.

## Single oocyst collection, visualization, and gDNA extraction

Between 11–12 days (*P. falciparum*) and 8–9 days (*P. yoelii*) post-infectious blood meal, the midguts of mosquitoes were dissected and stained with 0.1% Mercurochrome in PBS for 10 min. The midguts were briefly washed twice with PBS and fixed with 4% PFA for 5 min in order to avoid rupturing mature oocysts. Single oocysts from midguts infected with 1–25 oocysts were manually isolated using a 25 G needle, and isolated single oocysts were transferred to a 12-well glass slide containing 5 μL of 85% glycerol (Fisher Scientific) in PBS. Single oocysts were imaged using a phase microscope (Nikon, E600) and their diameter was measured using Fiji (https://fiji.sc/). After visualization, the single oocysts were transferred to a 1.5 ml tube and stored at -80 °C until genomic DNA (gDNA) extraction. gDNA from oocysts was extracted with a Monarch genomic DNA purification kit (New England Biolabs). Single oocysts were incubated with 200 μL of tissue lysis buffer and 10 μL of proteinase K for 30 min. gDNA extraction was conducted according to the manufacture's protocol and eluted in 50 μL. gDNA was stored at 4 °C for up to 1 week prior to sporozoite quantification by qPCR.

## Salivary gland collection and gDNA extraction

At the indicated days post-infectious blood meal, *P. yoelii*- and *P. falciparum*- salivary glands were dissected in PBS, transferred to a 1.5 ml tube containing 20 μL of PBS, and stored at -80 °C until gDNA extraction. gDNA was extracted by alkaline lysis. Salivary glands were lysed in 45 μL of 10 mM NaOH at 95 °C for 30 min and neutralized by adding 5 μL of 400 mM Tris-HCl (pH8.5) and 4 mM EDTA (pH8.5). gDNA was stored at 4 °C for up to 1 week prior to sporozoite quantification by qPCR.

## Ruptured oocyst quantification

For determination of the efficiency with which oocyst sporozoites colonize the salivary glands, midguts from the same mosquito used for salivary gland sporozoite quantification were dissected and immediately fixed with 4% PFA for 2 min under the stereo microscope stage. The fixed midguts were transferred to a 12-well glass slide containing 12 μL of 4% PFA and further incubated at room temperature for 1 h. After fixation, the midguts were permeabilized with 1% Triton-X 100 (Cepham Life Science) in PBS for 1 h, stained for CSP using 5 μg/ml mAb 2A10 for *P. falciparum*[45] or 5 μg/ml 2F6 for *P. yoelii*[46] in PBS with 5% BSA for 1 h at room temperature, followed by incubation with Alexa 488-conjugated anti-mouse antibody (Thermofisher, 1:500) in 1% BSA in PBS for 1 h at room temperature. The midguts were then washed with PBS three times and mounted with a gold antifade reagent with DAPI (Thermofisher). The ruptured oocysts were counted under the fluorescence microscope (Nikon, E600) and representative images were acquired by confocal microscopy (Zeiss LSM880).

## Single mosquito bite on mouse ear and gDNA extraction

Single mosquito feeds were performed as previously described[7]. On day 15-16 or 14-20 days post-infectious blood meal, *P. yoelii* and *P. falciparum*-infected mosquitoes, respectively, were anesthetized on ice and sorted into individual clear plastic 1 cm diameter tubes, which were capped with mesh netting at one end. After securing the open end with Parafilm, the mosquitoes were returned to the incubator and deprived of sugar water overnight. The next day, mice were lightly anesthetized by intraperitoneal injection of ketamine (35–100 μg/g) and xylazine (6–15 μg/g) and placed on a slide warmer maintained at 35 °C to prevent a drop in body temperature due to anesthesia. Because large amounts of mouse tissue could impact the detection of sporozoite DNA, each ear was taped to the slide warmer using standard lab tape with 6 mm hole punched out of the tape to

create a restricted window for mosquito probing. The tape was placed on the edge of the ear so that only half of the circle contained ear tissue; this functioned to both identify the area of probing and to geographically limit the probed tissue to an area that would facilitate downstream genomic DNA extraction. Since mosquito probing occasionally occurred at the periphery of the taped area, meaning that the area of interest would extend beyond the outline of the hole punched into the tape holding the mouse ear in place, a second piece of lab tape with a slightly larger 8 mm hole punch was used to assist in ear tissue collection, post-feed. The second piece of tape was placed on the underside of the ear so that both holes were in alignment and provided a larger outline of the probing area. For each experiment, a single plastic tube containing a starved mosquito was placed on the taped ear of the mouse and the mosquito was allowed access to the mouse ear, through the mesh, for 10–15 min. The duration of probing was recorded as the cumulative time that the mosquito proboscis was in the skin. Mosquitoes showing no initial interest in feeding, i.e., flying around the tube and not landing on the mouse for >5 min, were removed and the mouse was exposed to a replacement mosquito. The acquisition of a blood meal was determined by observing the abdomen of the mosquito for engorgement and red coloration, and confirmed by noting the presence of blood in the esophagus during later dissection using a stereo microscope. After completion of the feed, the mosquito was placed on ice, salivary glands were removed, and transferred to an individual 1.5 ml tube for gDNA extraction as outlined above. A single-use razor blade was used to collect ear tissue by cutting around the larger semi-circle of the second piece of tape, and then dividing the ear sample into two pieces that were approximately 10 mg or less, each of which was transferred to an individual 1.5 ml tube and stored on ice for no more than an hour prior to gDNA extraction. gDNA from mouse skin for sporozoite inoculum quantification was performed using the Monarch genomic DNA purification kit (New England Biolabs). Skin tissue was incubated with 200 μL of Tissue Lysis Buffer and 10 μL of proteinase for 2 h, with vortexing every 15 min. To remove any digested material that could clog the column, the samples were then centrifuged for 3 min at 15,000×*g* and the supernatants were transferred to a fresh 1.5 ml eppendorf tube for washing and elution of gDNA from the column, which was conducted using 50 μL of warmed elution buffer, according to the NEB protocol. Genomic DNA samples were stored at 4 °C for up to 1 month prior to qPCR quantification of sporozoites.

## Oocyst, salivary gland, and skin sporozoite quantification by qPCR

A standard curve was made by isolating gDNA from known numbers of salivary gland sporozoites, counted using a haemocytometer. A serial dilution of the sporozoites was performed to yield a log-fold dilution of sporozoites from 20 to $2 \times 10^5$ (sometimes $5 \times 10^5$). Aliquots of known numbers of sporozoites were frozen at −80 °C until they were processed with the experimental samples, as described above. gDNA for sporozoite standards were obtained using both alkaline lysis (for salivary gland sporozoites) and the Monarch® gDNA extraction kit (for skin sporozoites), and an appropriate set of standards was run on each reaction plate. qPCR of salivary gland or single oocyst sporozoites was performed with LSUE primers (forward primer, 5′-CGG TCC TAA GGT AGC AAA ATT CCT-3′ and reverse primer 5′-AGG AGT CTC ACA CTA GCG ACA ATG-3′). LSUE is a 92 base pair amplicon (5′-CGG TCC TAA GGT AGC AAA ATT CCT TGT CGG GTA ATC TCC GTC CTG CAT GAA CGG TGT AAC GAC TTC CCC ATT GTC GCT AGT GTG AGA CTC CT-3′) contained in a mitochondrial rRNA fragment (PF3D7_0214200), and is believed to encode a fragment of a rRNA large subunit[47–49]. The sequence for this amplicon is conserved between *P. berghei, P. yoelii*, and *P. falciparum* (www.plasmodb.org). *Plasmodium falciparum* salivary gland sporozoite quantification in Fig. 2 utilized 18 s rRNA primers

(forward primer, 5′- CCT GGT TGA TCT TGC CAG TAG-3′ reverse primer 5′-ATG AGC CGT TCG CAG TTT-3′), and 4 μL gDNA in a total volume of 20 μL/well were used and qPCR was performed with the StepOnePlus™ system (Applied Biosystems) using either GoTaq®qPCR master mix (Promega) or SYBR® green qPCR master mix (Thermofisher). The cycling profile for GoTaq®qPCR master mix (Promega) was 95 °C for 2 min followed by 40 cycles of: 95 °C, 3 s; 60 °C, 30 s. The cycling profile for SYBR® green qPCR master mix (Thermofisher) was 95 °C for 10 min followed by 40 cycles of: 95 °C, 15 s; 63.5 °C, 60 s. After amplification, the melting temperature was determined using a dissociation curve to ensure that a single, specific product was formed. The profile for the melt curve was 95 °C for 15 s, 60 °C for 60 s, and incremental increases of 3 °C up to 95 °C.

For gDNA samples extracted from mouse ear, a probe-based assay was used to detect LSUE transcripts with the forward and reverse primers previously described and a novel LSUE probe (Express PrimeTime 5′ 6-FAM/ZEN/3′ IBFQ Probe, 5′ 6-FAM/TCG GGT AAT/ZEN/CTC CGT CCT GCA TGA AC/3′ IABkFQ, IDT). Each reaction totaled 20 μL, including 10 μL of PrimeTime® Gene Expression Master Mix (IDT), 1.5 μL each of 5 μM forward primer, reverse primer, and probe, and 3 μL of template. All runs were conducted for 40 cycles with the following cycling parameters: initial denaturation: 95 °C for 3 min, followed by 40 cycles of denaturation at 95 °C for 5 s and annealing, extension, and detection of fluorescence at 55 °C for 30 s.

### Visualization of the mosquito bite site

Single mosquito feeds were performed as described above. The mouse ears were immediately fixed in 4% PFA on ice for 1 h, divided into dorsal and ventral sheets, and further fixed in 4% PFA at room temperature for 3 h. Following this, ears were incubated at 4 °C overnight in 4% PFA with 0.3% Triton-X100 and 5% goat serum in PBS. Ear leaves were then immersed in a 50% sucrose solution for 24 h at 4 °C, followed by freezing on dry ice and stored at −80 °C for 48 h. Upon thawing to room temperature, they were stained with mAb 2A10 conjugated with Alexa488 in a 5% goat serum in PBS for 4 h, with shaking at 4 °C. Ear leaves were washed twice with 5% goat serum in PBS with shaking for 15 min and mounted onto a 2-well glass slide with a gold antifade reagent. The sporozoites were located at the mosquito bite site and imaged using a Zeiss LSM880 confocal microscope. Quantification was performed using Image J.

### Statistical analysis

All data were tested for normality, followed by the appropriate statistical test in Graphpad Prism (Version 7 or 8.4). $\chi^2$ tests were used to test associations between frequency distributions of salivary gland sporozoite loads and inoculum size. Odds ratios and 95% confidence intervals were calculated using the exact method[50].

### Reporting summary

Further information on research design is available in the Nature Portfolio Reporting Summary linked to this article.

## Data availability

All of the raw data used to make the main and supplementary figures are included in the Source Data excel file. Source data are provided with this paper.

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

## Acknowledgements

We would like to thank Godfree Mlambo, Abhai Tripathi, and Chris Kizito, the parasitology and insectary core facilities team at the Johns Hopkins Malaria Research Institute, and the Johns Hopkins School of Medicine Microscopy Facility (MicFac). This work was supported by a Johns Hopkins Malaria Research Institute fellowship to S.K., Bloomberg Philanthropies and the National Institutes of Health (R01AI132359 to P.S. and T32AI138953 to D.S.), NIH grants that funded the MicFac LSM880 (S10OD023548) and the European Research Council (ERC-QUANTUM to T.B.). We would also like to acknowledge statistical support from the Johns Hopkins Institute for Clinical and Translational Research, which is funded by the National Center for Advancing Translational Sciences of the National Institutes of Health (UL1TR003098).

## Author contributions

S.K. and D.S. performed all of the experiments and analyzed the data. P.S. supervised the project and contributed to data analysis. G.Y. performed statistical analysis of the data, and T.B. contributed to project development.

## Competing interests

The authors declare no competing interests.
