## [Peer Review File · Nature Communications]

Revisiting the Plasmodium sporozoite inoculum and elucidating the efficiency with which malaria parasites progress through the mosquitoREVIEWER COMMENTS

Reviewer #1 (Remarks to the Author):

Overall impression: This study was well designed, well executed, and the manuscript is well written.

- What are the noteworthy results?

Noteworthy results center on their finding that *Plasmodium falciparum*, but not *Plasmodium yoelii*, inoculation rates are higher in a subset of highly SG-infected *Anopheles* mosquitoes.

- Will the work be of significance to the field and related fields?

Yes. This is important information for further refinement of epidemiological models and evaluations of transmission-blocking interventions.

- How does it compare to the established literature?

The current work aligns well with previously published literature. The authors make an effort to properly place their work within the context of previous studies.

- If the work is not original, please provide relevant references.

The work included in the current study is original.

- Does the work support the conclusions and claims, or is additional evidence needed?

The work completed narrowly supports the conclusions presented. There is more work to be done regarding the role of co-evolution, salivary gland cell biology, and

- Are there any flaws in the data analysis, interpretation and conclusions?

There are no obvious flaws in the data analysis, interpretation, and conclusions. Statistical testing was carried out appropriately. The limitations of the study are included within the Discussion, although not directly stated.

- Do these flaws prohibit publication or require revision?

No.

- Is the methodology sound?

Yes.

- Does the work meet the expected standards in your field?

Yes.

- Is there enough detail provided in the methods for the work to be reproduced?

Yes.

Questions for the research team:

1) What role could variability in individual salivary gland cell biology, as well as between salivary glands of different mosquito species, play in infectivity? Do salivary gland parasites face barriers which could impact inoculation, or could be influenced by sporozoite size (e.g. PMID: 31387905)?

Alternatively, is there any evidence of a molecular lock-and-key system involved in initiating inoculation, similar to immune evasion in the midgut (e.g. PMID: 31969456)? Could additional factors like sporozoite disintegration influence counts of inoculated sporozoites (e.g. PMID: 35403820; how does the timing of inoculation count assays compare)?

2) Purely based on mosquito salivary gland volume and sporozoite volumes, how many sporozoites can a mosquito salivary gland hold before stretching and/or rupture might occur? How does this number relate to the 10,000 sporozoite threshold for inoculation that you describe?

3) You've addressed the impact of evolutionary and co-evolutionary differences between *Plasmodium falciparum* and *Plasmodium yoelii* very well. However, this opens additional questions related to co-evolution. *Plasmodium berghei* is a second, highly utilized rodent malaria species in laboratory research studies. How large are *Plasmodium berghei* sporozoites, and what do available data suggest the relationship between mosquito midgut/salivary gland infection and infectivity to be in this parasite species?

4) What is going on in the subset of highly inoculating mosquitoes after sporozoites leave the salivary glands? This could be related to the current study, or analyzed, in greater detail using previous studies or methods (e.g. PMID: 15186404).

Reviewer #2 (Remarks to the Author):

This is an important study that uses human (*Plasmodium falciparum*) and rodent (*P. yoelii*) malaria parasite species in the Indo-Pakistan vector mosquito, *Anopheles stephensi*, to describe quantitatively the impact of the intensity of infections of mosquito-stage malaria parasites (oocysts and sporozoites) on the final inoculum delivered to the vertebrate host. These are important numbers to have, and they set the groundwork for answering the final, important and as-of-yet unanswered question of how many sporozoites does it take to establish an infection in the host. I look forward to the type of efforts described by the authors in lines 445-448 that will give us that answer. I also agree with the author statements (lines 389-393) that it will be important to carry out similar studies with *An. gambiae* and *An. coluzzi*, the primary vectors in sub-Saharan Africa to determine how broadly their findings can be applied to other transmission scenarios (vectors and parasite species).

The experiments were well-designed, the outcomes unequivocal and the conclusions justified. The experimental protocols also provide a good example to others on how to carry out similar experiments.

I have returned marked copies of both the manuscript, figures and supplementary information. The authors should reference these for editorial suggestions and corrections. I also summarized below the reason for some of the suggested changes.

Line in manuscript Comment

21 Keywords should not appear in the title

34 Malaria is the disease; parasites are transmitted

34 anopheline is by convention lower-case

45 Give some idea of what 'high inocula' means, for example, >10,000spz?

98 and throughout 'Though' is vernacular the way it is used here, 'Although' is more formal.

199 and throughout Spell out all genus names when they start sentences. This avoids having one-letter length sentences.

120 and throughout 'load' is used to mean the intensity of infection. It should be defined as such and the two terms then can be used interchangeably throughout the text.

136 Insert a couple of sentences describing how the mosquitoes were infected. This will help the logic flow between the preceding and following paragraphs.

239 See note on edited ms about the usage of 'fecundity'

255 Not clear what is meant by 'cages', collections of individual mosquitoes?

AAJ 20231106

Reviewer #3 (Remarks to the Author):

This is an outstanding work. They cultured *P. falciparum* gametocytes and prepared infectious mosquitoes. Then they dissected the mosquito salivary glands one by one, and estimated the number of sporozoites using qPCR. They also use numerous numbers of mice to be bitten by the infectious mosquitoes. Each mouse skin was removed and extracted for parasite quantification. Finally, they presented outstanding volume of results. I was surprised that they did these works without using transgenic malaria parasites.

Most impressive result was Figure 2. They concluded that 3 oocysts were sufficient to prepare a *P. falciparum* infectious mosquito. Except for this result, other results were not so exciting for me.

I would like to pay tribute to their hard work. I recognize that this work will be one of the historic milestones in malaria research.

Please correct some words.

L70: exo-erythrocytic stage (EEF) > exo-erythrocytic form (EEF)?

L87: wcould > would or could?

We appreciate and thank the reviewers for their comments and suggestions. Below are our responses to individual reviewers comments.

Reviewer #1

Questions for the research team:

1) What role could variability in individual salivary gland cell biology, as well as between salivary glands of different mosquito species, play in infectivity? Do salivary gland parasites face barriers which could impact inoculation, or could be influenced by sporozoite size (e.g. PMID: 31387905)?

Alternatively, is there any evidence of a molecular lock-and-key system involved in initiating inoculation, similar to immune evasion in the midgut (e.g. PMID: 31969456)?

We are answering the above questions together because they share some important features. In summary they are interesting and important questions about which there are little to no published data. To date, we know of no studies comparing salivary gland cell biology among individual mosquitoes and in different species. It is very possible that differences in salivary gland biology/architecture could impact expelling and this would be very interesting to follow up on, yet is beyond the scope of this study. We agree that parasites face barriers as was nicely shown in the Andrews paper (PMID: 31387905) and there could be differences among mosquitoes that could be important. Specifically, in a previous publication from our group (PMID: **15972531**) we found that a small group of mosquitoes repeatedly inoculated no sporozoites – these would be very interesting to study! For this question as well as question 4 below, it would be informative to visualize the process of sporozoite mobilization and movement into the salivary duct in real time to determine if sporozoites in different parts of the gland are successful. We have been trying to do this for many years but the cuticle of the mosquito impedes access. Freddy Frischknecht in his Cell Micro paper could only visualize the duct and our unpublished data suggest similar to his data, *P. falciparum* sporozoites are rarely in the ducts and when there are in single file. So its clear that the secretory cavities of mosquitoes need to be visualized during probing – this will be an important contribution. We know of no evidence for a lock and key mechanism for either salivary gland invasion or the initiation of inoculation. However, mosquitoes do have host preferences, which according to our colleague Conor McMeniman, are based on scent blends. We imagine that once a mosquito is attracted to its host, that inoculation proceeds – no data to suggest otherwise but interesting to consider.

Could additional factors like sporozoite disintegration influence counts of inoculated sporozoites (e.g. PMID: 35403820; how does the timing of inoculation count assays compare)?

Good question since as soon as sporozoites are inoculated they begin to move and be seen by the immune system, etc. Interestingly, in our time-lapse imaging of wildtype sporozoites, *P. berghei*, *P. yoelii*, and *P. falciparum* (PMID: 26271010 and PMID: 33750026) we never saw destruction of sporozoites or disintegration over the imaged 2 hr time frame. In the current study, we gave the mosquitoes 10 to 15 minutes to probe and as soon as the mosquito was removed the ear was removed and put on ice, which should stop sporozoite motility and any immune processes. Thus, in the time frame of our analysis, a maximum of ~10 minutes post-inoculation, we think we captured the majority of sporozoites prior to any destruction.

2) Purely based on mosquito salivary gland volume and sporozoite volumes, how many sporozoites can a mosquito salivary gland hold before stretching and/or rupture might occur? How does this number relate to the 10,000 sporozoite threshold for inoculation that you describe?

There are no published data on this topic but our individual salivary gland sporozoite load data (see excel sheet included with our revised paper and Figure 1) clearly demonstrate that mosquitoes can harbor over 100,000 salivary gland sporozoites. In the laboratory it is not difficult to obtain such numbers, however, in the field these mosquitoes exist but are rare (PMID: 1768903, PMID: 9383768, PMID: **6006329**), raising the possibility that they are super-spreading mosquitoes. So the 10 to 20K threshold is well below what a salivary gland is able to ‘hold’ at least for *P. falciparum*.

3) You’ve addressed the impact of evolutionary and co-evolutionary differences between Plasmodium falciparum and Plasmodium yoelii very well. However, this opens additional questions related to co-evolution. Plasmodium berghei is a second, highly utilized rodent malaria species in laboratory research studies. How large are Plasmodium berghei sporozoites, and what do available data suggest the relationship between mosquito midgut/salivary gland infection and infectivity to be in this parasite species?

We agree that given the wide use of *P. berghei* these numbers could be helpful to the field. We have actually done a separate study of *P. berghei* and the numbers are similar to what we observe in *P. yoelii*. We did not put these data in this paper as we wanted to minimize the distraction from *P. falciparum*, the main topic. However we will submit this for publication soon and copy some of the data below in case its helpful to you.

Though

- A) Infection likelihood after single infected bite with mosquito having < or > 10K sporozoites.
- B) Sporozoite inoculum as a function of salivary gland sporozoite load – *P. yoelii* and *P. berghei* data shown.
- C) Salivary gland sporozoite load as a function of ruptured oocyst number – *P. yoelii* and *P. berghei* data shown.

the data for *P. berghei* are similar to *P. yoelii*, it does appear that salivary gland entry by oocyst sporozoites is somewhat more efficient by *P. berghei* (panel C). Nonetheless, it still requires over 15 oocysts to reliably reach salivary gland loads >10K, while for *P. yoelii* its >20 oocysts.

Regarding sporozoite length, we have not measured *P. berghei* oocyst sporozoite length, which is what we show in the paper. However oocyst sporozoite length is not very different from salivary gland sporozoite length and below we show a comparison of *P. berghei*, *P. yoelii*, and *P. falciparum* salivary gland sporozoite length. *P. berghei* sporozoites are longer than *P. falciparum* but not quite as long as *P. yoelii*.

4) What is going on in the subset of highly inoculating mosquitoes after sporozoites leave the salivary glands? This could be related to the current study, or analyzed, in greater detail using previous studies or methods (e.g. PMID: 15186404).

We assume the reviewer is asking about what happens in the skin and beyond. This is an important question as its possible that sporozoites from highly infected mosquitoes are different, possibly more infectious. These experiments are currently underway in the laboratory.

Reviewer #2 (Remarks to the Author):

This is an important study that uses human (*Plasmodium falciparum*) and rodent (*P. yoelii*) malaria parasite species in the Indo-Pakistan vector mosquito, *Anopheles stephensi*, to describe quantitatively the impact of the intensity of infections of mosquito-stage malaria parasites (oocysts and sporozoites) on the final inoculum delivered to the vertebrate host. These are important numbers to have, and they set the groundwork for answering the final, important and as-of-yet unanswered question of how many sporozoites does it take to establish an infection in the host. I look forward to the type of efforts described by the authors in lines 445-448 that will give is that answer. I also agree with the author statements (lines 389-393) that it will be important to carry out similar studies with *An. gambiae* and *An. coluzzi*, the primary vectors in sub-Saharan Africa to determine how broadly their findings can be applied to other transmission scenarios (vectors and parasite species).

The experiments were well-designed, the outcomes unequivocal and the conclusions justified. The experimental protocols also provide a good example to others on how to carry out similar experiments.

I have returned marked copies of both the manuscript, figures and supplementary information. The authors should reference these for editorial suggestions and corrections. I also summarized below the reason for some of the suggested changes.

Thank-you for carefully going through the manuscript. We have incorporated all of your suggested edits with the exception of line 350 where we changed “connect” to “link” rather than “index”.

Line in manuscript Comment

21 Keywords should not appear in the title

We have now removed these keywords

34 Malaria is the disease; parasites are transmitted
Agree and changed!

34 anopheline is by convention lower-case
now changed in all instances where anopheline is used

45 Give some idea of what 'high inocula' means, for example, >10,000spz?
Done but since the Abstract had to be shortened this was deleted.

98 and throughout 'Though' is vernacular the way it is used here, 'Although' is more formal.
Now changed throughout the manuscript

199 and throughout Spell out all genus names when they start sentences. This avoids having one-letter length sentences. Done!

120 and throughout 'load' is used to mean the intensity of infection. It should be defined as such and the two terms then can be used interchangeably throughout the text.
Good point - now defined in line 111, the first use in the paper (excluding the abstract).

136 Insert a couple of sentences describing how the mosquitoes were infected. This will help the logic flow between the preceding and following paragraphs.
We agree that the logical flow was not there and think it was because we put the generation of mosquito cages and qPCR quantification in the same paragraph. We've now added some text to the generation of mosquito cages and moved the sentences on quantification of inoculum to the paragraph below.

239 See note on edited ms about the usage of 'fecundity'
Now defined at first use

255 Not clear what is meant by 'cages', collections of individual mosquitoes?
now changed to 'batches' – does this make better sense?

AAJ 20231106

Reviewer #3 (Remarks to the Author):

This is an outstanding work. They cultured *P. falciparum* gametocytes and prepared infectious mosquitoes. Then they dissected the mosquito salivary glands one by one, and estimated the number of sporozoites using qPCR. They also use numerous numbers of mice to be bitten by the infectious mosquitoes. Each mouse skin was removed and extracted for parasite quantification. Finally, they presented outstanding volume of results. I was surprised that they did these works without using transgenic malaria parasites.

Most impressive result was Figure 2. They concluded that 3 oocysts were sufficient to prepare a *P. falciparum* infectious mosquito. Except for this result, other results were not so exciting for me.

I would like to pay tribute to their hard work. I recognize that this work will be one of the historic milestones in malaria research.

Please correct some words.

L70: exo-erythrocytic stage (EEF) > exo-erythrocytic form (EEF)?

Changed to **exo-erythrocytic form**

L87: wcould > would or could?

Changed to **"would"**